# PKCα Activation via the Thyroid Hormone Membrane Receptor Is Key to Thyroid Cancer Growth

**DOI:** 10.3390/ijms252212158

**Published:** 2024-11-13

**Authors:** Mateo N. Campos Haedo, Johanna A. Díaz Albuja, Sandra Camarero, Florencia Cayrol, Helena A. Sterle, María M. Debernardi, Marina Perona, Melina Saban, Glenda Ernst, Julián Mendez, María A. Paulazo, Guillermo J. Juvenal, María C. Díaz Flaqué, Graciela A. Cremaschi, Cinthia Rosemblit

**Affiliations:** 1Instituto de Investigaciones Biomédicas (BIOMED), Consejo Nacional de Investigaciones Científicas y Técnicas (CONICET), Facultad de Ciencias Médicas, Pontificia Universidad Católica Argentina (UCA), Buenos Aires C1107AFB, Argentina; mateocamposhaedo@gmail.com (M.N.C.H.); johanna_diazalbuja@uca.edu.ar (J.A.D.A.); florencia_cayrol@uca.edu.ar (F.C.); helena_sterle@uca.edu.ar (H.A.S.); mdebernardi@uca.edu.ar (M.M.D.); alejandra_paulazo@uca.edu.ar (M.A.P.); diazflaque@gmail.com (M.C.D.F.); 2Histopathology Service, Hospital de Pediatría Garrahan, Buenos Aires C1245AAM, Argentina; sandracamarero2@gmail.com; 3Departamento de Radiobiología, Comisión Nacional de Energía Atómica (CNEA), Buenos Aires B1650KNA, Argentina; mperona@cnea.gov.ar (M.P.); juvenal@cnea.gov.ar (G.J.J.); 4Consejo Nacional de Investigaciones Científicas y Técnicas (CONICET), Buenos Aires C1425FQB, Argentina; 5Endocrinology Service, Hospital Británico de Buenos Aires, Buenos Aires C1280AEB, Argentina; msaban@hbritanico.com.ar; 6Scientific Committee, Hospital Británico de Buenos Aires, Buenos Aires C1280AEB, Argentina; GErnst@hbritanico.com.ar; 7Histopathology Service, Hospital Británico de Buenos Aires, Buenos Aires C1280AEB, Argentina; jumendez@hbritanico.com.ar

**Keywords:** PKCα, thyroid hormones (THs), integrin αvβ3, thyroid cancer (TC)

## Abstract

Thyroid carcinoma (TC) is the most common endocrine neoplasia, with its incidence increasing in the last 40 years worldwide. The determination of genetic and/or protein markers for thyroid carcinoma could increase diagnostic precision. Accumulated evidence shows that Protein kinase C alpha (PKCα) contributes to tumorigenesis and therapy resistance in cancer. However, the role of PKCα in TC remains poorly studied. Our group and others have demonstrated that PKCs can mediate the proliferative effects of thyroid hormones (THs) through their membrane receptor, the integrin αvβ3, in several cancer types. We found that PKCα is overexpressed in TC cell lines, and it also appeared as the predominant expressed isoform in public databases of TC patients. PKCα-depleted cells significantly reduced THs-induced proliferation, mediated by the integrin αvβ3 receptor, through AKT and Erk activation. In databases of TC patients, higher PKCα expression was associated with lower overall survival. Further analyses showed a positive correlation between PKCα and genes from the MAPK and PI3K-Akt pathways. Finally, immunohistochemical analysis showed abnormal upregulation of PKCα in human thyroid tumors. Our findings establish a potential role for PKCα in the control of hormone-induced proliferation that can be explored as a therapeutic and/or diagnostic target for TC.

## 1. Introduction

Thyroid cancer (TC) incidence has significantly increased within the last five decades worldwide, becoming the most common endocrine system malignancy [1,2,3]. Thyroid tumors are subdivided by histological subtypes. Differentiated TC (DTC) of follicular origin, which are the most common forms of TC including papillary (PTC) and follicular (FTC) TC, make up about 85–95% of all cases. There are rarer types like oncocytic carcinoma (OCA) and poorly differentiated TC (PDTC), accounting for 2–5% of cases, and anaplastic TC (ATC), found in about 1.7% of all cases. Medullary TC (MTC) is less common, making up 3–5% of cases [4,5,6,7]. Localized DTC has a 95% survival rate at 5 years and is associated with a highly favorable prognosis [2,8]. PDTC and ATC patients have the worst prognosis, and a lower overall survival (OS) rate with a mean survival of about 3.2 years and 6 months, respectively [9,10]. Although first-line treatment for DTC has been surgery, it is associated with potential morbidity (e.g., need for TH treatment, hypoparathyroidism, and recurrent laryngeal nerve injury). Active surveillance has been proposed as an alternative for small low-risk DTC, which means close monitoring of the primary cancer without performing an initial surgery or other intensive treatments [2,11,12,13]. In addition, active surveillance and scrutiny of the thyroid gland have led to the increased detection of small, early-stage tumors but they might also lead to the discovery of some clinically silent, larger, and advanced-stage cancers, increasing the number of advanced-stage cancers reported in some countries [14,15,16]. Thus, it is necessary to find therapies that contribute to improving the quality of life and survival of patients, or to optimizing existing therapies to reduce their adverse effects. Despite its increasing incidence and mortality in many cases, TC constitutes a poorly studied area when compared to other cancer types.

Protein kinase C (PKC) represents a family of 10 different serine/threonine kinases that regulate proliferation, apoptosis, cell survival, migration and tumorigenesis [17]. The different isoforms have been classified into three groups according to their different regulation: (1) classical or calcium-dependent (PKCα, βI, and βII y γ); (2) novel or calcium-independent (PKCδ, ε, η, and y θ); and (3) atypical (PKCs ζ and ι). Classical and novel isoforms are activated by diacylglycerol (DAG), a product of the hydrolysis of phospholipids by the action of phospholipase C (PLC), which is activated in response to tyrosine kinase receptors and G protein-coupled receptors (GPCRs). Phosphatidylserine can regulate the activity of the three groups [18,19,20]. Understanding how PKC signal deregulation contributes to tumorigenesis has not been easy, as each individual PKC plays different roles in different cell types [21]. Classical PKCs participate in the majority of cellular signaling pathways and thus any aberration in their activity leads to the development of pathological conditions such as cancer, among others [18,20,22].

PKCα signaling has been implicated in the regulation of cell proliferation, differentiation, cell survival, and tumorigenesis, as well as cell motility, migration, invasion, and metastasis in cancer cells that can be context-dependent [23]. PKCα overexpression is a predictor of a poor prognosis and drug resistance in breast [24,25,26,27,28,29]; lung [30,31,32]; prostate [33]; ovarian [34,35,36]; and hepatocellular cancer [37]; and melanoma [38,39,40] and glioma [41,42,43], among others. Therefore, PKCα has been proposed as a biomarker of poor prognosis and resistance to therapy in patients with several cancers. In general, PKCα protects cancer cells against apoptosis, triggering a survival response [20]. However, in the intestine, the overexpression of PKCα is a predictor of a good prognosis as it is involved in cell death pathways [23,44,45].

Our group has demonstrated that PKCζ participates in T-cell lymphoma (TCL) cell proliferation induced by thyroid hormones (THs), 3,5,3′-triiodo-L-thyronine (T3), and L-thyroxine (T4) through complementary intracellular pathways involving nuclear (TRs) and membrane receptors (mTRs). By binding to its mTR, the integrin αvβ3, THs cause the phosphorylation of Erk in TCL cells, which in turn activates the transcription factor NF-κB, involved in cell proliferation, angiogenesis, and drug metabolism [46,47,48,49,50]. In TC, THs, acting via the integrin αvβ3 receptor lead to MAPK/ERK1/2 activation, promoting cell proliferation and inhibiting apoptosis [51]. Thus, in addition to their normal action on cells and tissue metabolism, THs have been implicated in cell transformation, tumorigenesis, angiogenesis, and metastatic spread.

To improve the diagnosis and treatment of TC, a greater study of tumorigenesis and the molecular bases that lead to the dysregulation of thyroid cell growth is crucial. In this study, we sought to determine whether PKCα regulates signaling pathways involved in the control of TC proliferation by THs through their membrane receptor (mTR), the integrin αvβ3. Also, we evaluated the expression and prognostic value of PKCα in TC patients from public databases and from an Argentine hospital. Therefore, we hypothesized that a therapy targeted to decrease PKCα expression, or its biological activity would contribute to the tumor pathology remission. Finally, we hypothesized that the detection of PKCα expression could have a prognostic value in patients with this type of malignancy.

## 2. Results

### 2.1. PKCα Is Overexpressed in TC Cells

As mentioned, the change in the expression pattern of protein kinase C (PKC) isoforms has been associated with the development of different types of cancer [21,52]. Here, we first tested the mRNA expression of different members of the PKC family in human-immortalized normal and thyroid cancer (TC) cell lines. The mRNA of conventional, novel, and atypical PKC isoenzymes, shown to be involved in cancer development and progression, were found in the studied TC cell lines. Also, qPCR assays revealed high levels of PKCα mRNA in most TC cells (Figure 1A). Western blot assays of PKCα also showed higher levels of this isoenzyme relative to the nontumorigenic Nthy-ori-3-1 (ORI) thyroid cells in multiple TC cell lines (Figure 1B). The protein levels of some PKC family members were also present in the different types of TC cell lines. Additionally, PKCγ and PKCδ were the only PKC isoforms mainly expressed in the anaplastic TC cell line 8505C (Appendix A).

### 2.2. PKCα Is the Predominant Expressed Isoform in TC Patients

To verify our in vitro results, we performed analysis on a public patients’ dataset through the R2: Genomics Analysis and Visualization Platform (GSE126729) [53]. First, we analyzed the mRNA levels of all the members of the PKC family in TC patients (papillary PTC, follicular FTC, and anaplastic ATC) and we found that the gene PKCα is highly expressed in all TC with respect to the genes of other isoforms (Figure 2A). Similarly, when we repeated the analysis only in ATC patients, we found that PKCα is the isoform with the highest mRNA expression level (Figure 2B).

We also evaluated PKCα staining by immunohistochemistry (IHC) using an anti-PKCα antibody previously validated [54], in specimens from TC patients. We were not able to collect samples from ATC patients, but we obtained tissue from nine papillary microcarcinoma patients. We observed heterogeneous PKCα staining across different specimens, with cases displaying very high staining and others with weak or no staining. Only neoplasms that expressed PKCα intensely and diffusely in a cytoplasmic pattern were considered positive. The rest of them were considered negative. It is to be noted that negative samples display high levels of PKCα positive lymphoid infiltration (arrow in Figure 2C, lower panel). Representative examples are shown in Figure 2C. A detailed analysis revealed low- or no-intensity staining in non-tumor areas, whereas PKCα staining was primarily detected in those areas defined as having adenocarcinoma. An enhanced view is shown in the Figure 2C inset. Based on the available clinical information, the results of our study suggest an increased level of PKCα in 33% of patients with papillary thyroid microcarcinoma.

### 2.3. Relationship Between PKCα Expression Levels and Overall Survival in TC Patients

To evaluate the impact of PKCα gene levels in TC patients, we studied the survival of TC patients with low or high *PRKCA* expression. Kaplan–Meier curves for male TC patients expressing high levels of *PRKCA* were significantly correlated with poor overall survival (OS) in the TCGA-PanCancer Atlas dataset (*p* < 0.001) (Figure 3A, left panel), and the same tendency was found in female TC patients with high *PRKCA* expression levels (Figure 3A, right panel). Also, we studied the relationship of tumor mutational burden (TMB) with survival in TC patients with low and high *PRKCA* expression. TMB, widely considered a prognostic marker for immunotherapy, has been previously associated with worse overall survival and is positively associated with recurrence in TC [55,56]. Independently of TMB, patients with high *PRKCA* levels have a reduced OS than those with low expression (*p* < 0.01) (Figure 3B). Interestingly, TC patients with high TMB and high *PRKCA* expression have also a reduced OS than those with low expression (*p* < 0.05) (Figure 3B, right panel). These results indicate that PKCα expression could have a role in reducing TC patients’ survival.

### 2.4. PKCα Expression Is Associated with Proliferation Pathways in TC Patients

Given the decrease in OS of TC patients with high PKCα expression levels, we then performed in silico analysis in cBioPortal on the PanCancer Atlas TC dataset that showed a positive correlation between PKCα and MAPK4 (S = 0.46, *p* < 0.001) (Figure 4A) and PIK3CG (S = 0.35, *p* < 0.001) from the PI3K-AKT pathway (Figure 4B). Moreover, enrichment analysis through the Metascape platform showed an induction of MAPK and PI3K-AKT programs in samples where PKCα is overexpressed (Figure 4).

### 2.5. PKCα Mediates Hormone-Induced Proliferation in TC

To obtain representative results for TC, we chose cell lines from the different cancer types overexpressing PKCα. Thus, papillary TPC-1, follicular WRO, and anaplastic 8505C were incubated with 3,5,3′-triiodo-L-thyronine (T3) and L-thyroxine (T4) alone or in combination for 48 h, and proliferation was evaluated. Both T3 and T4, in physiological concentrations, lead to cell proliferation in all cell lines, with the greatest increase when T3 and T4 were added together as found in circulation (Appendix A).

To explore the role of PKC in hormone-induced proliferation in TC, we performed Cell Titer Blue assays in the presence or absence of PKC inhibitors staurosporine (STAU) and GF109203X (GF) with hormones for 48 h (Figure 5A). STAU is a competitive PKC inhibitor with high binding affinity and low specificity [20,57]. On another hand, we used the pan-PKC inhibitor GF, which preferentially inhibits classic PKCs and is a selective inhibitor with a ranked order of potency (α > β l > ε > δ > ζ) [58,59,60]. We found that both inhibitors diminish THs proliferation in TC cells, indicating PKC participation (Figure 5A). As expected, PKC inhibition also reduces thyroid stimulating hormone (TSH)-induced proliferation, used as the positive control (Figure 5A). Similar results were obtained in WRO and 8505C cells (Appendix A). Since PKCα is the most upregulated classical PKC in our TC cell lines, it is likely that this PKC isoenzyme mediates TSH- and THs-induced proliferation.

To confirm PKCα participation in hormone-induced proliferation, we knocked-down PKCα using siRNA. A depletion of >80% of PKCα was achieved (Figure 5B, inset). As shown in Figure 5B, TSH- as well as TH-induced proliferation was inhibited in PKCα-depleted cells compared to cells transfected with a human non-coding sequence (siNT). In this sense, both TSH and THs increase PCNA expression levels compared to untreated cells (Figure 5C). Since PCNA is a proliferation marker molecule, this is consistent with our previous result demonstrating PKCα’s knocked-down effect on hormone-induced proliferation; PCNA expression levels were diminished in PKCα-depleted cells treated with TSH and THs compared to siNT cells treated with hormones (Figure 5C). In line with these results, we found increased apoptosis in PKCα-depleted cells demonstrated by increased active caspase-3 expression compared to control cells (Figure 5C). In summary, our results suggest not only that PKCα mediates TSH- and THs-induced proliferation, but also plays an important role for PKCα in the inhibition of apoptosis in TC cells.

To validate the in silico findings, we studied AKT and p42/p44 MAPK activation by PKCα. We found that TSH and THs treatment for 10 min induced AKT phosphorylation on serine 473 and p42/p44 MAPK phosphorylation in threonine 202 and tyrosine 204 residues, as shown by Western blot analysis of TPC-1 cells (Figure 5D). PKCα knock-down impaired both AKT and p42/p44 MAPK activation induced by TSH and THs (Figure 5D), clearly indicating that this pathway is carried out by PKCα signaling.

### 2.6. Thyroid Cells Express Integrin αvβ3, THs Membrane Receptor

Integrin αvβ3 levels were demonstrated to correlate with increased proliferation, invasiveness, and metastasis in different types of cancer [49,61,62] and PKCα was suggested to participate in the downstream signaling pathway triggered by αvβ3 in melanoma [61] and breast cancer [63].

We found that all the cell lines studied express αv and β3 by qPCR (Appendix A). Also, by flow cytometry analysis, we confirmed the presence of the dimer in the cell lines tested (Appendix A).

To more deeply evaluate the potential role of PKCα in the poor outcome of TC patients, we analyzed its molecular participation in the signaling pathways triggered via the integrin αVβ3 activation by THs [64,65].

We then performed a bioinformatic analysis with the cBioPortal program [66,67] in patients with different types of cancer we verified the existence of the genes ITGAV and ITGB3 (mTR) and found that thyroid tumors are the ones with the highest αv and β3 integrin expression (Appendix A).

To study the participation of integrin αvβ3 in THs-induced proliferation, cells were incubated with the specific RGD peptidomimetic inhibitor Cilengitide [68]. Dose–response curves of Cilengitide were plotted in the selected cell lines (Appendix A), and 4.5 μM of Cilengitide (the dose usually used for in vivo assays) [49] inhibited THs-induced proliferation in TPC-1, WRO, and 8505C cells (Figure 6), demonstrating the role of the integrin αvβ3. As αvβ5 and α5β1 were also reported as Cilengitide targets in cancer [68,69,70,71], we performed another bioinformatic analysis through cBioPortal in patients with different types of cancer and verified the genes that encode these other integrins: α5, β1, and β5, are not expressed in CT patients (Appendix A).

Bioinformatics analyses in TC patients support our results in vitro, demonstrating that PKCα is the predominant PKC isoform in TC with a leading role in proliferation.

## 3. Discussion

Protein Kinase C alpha (PKCα) a is here remarked as an important marker of thyroid cancer (TC) progression as its expression levels are related to worse clinical and tumor parameters.

We found that PKCα is the main PKC isozyme overexpressed in human cell lines of the different types of TC. PKCα overexpression has been described in lung, breast, and prostate cancer, and high-grade glioma and melanoma, among others [24,30,33,35,38,41], and it is generally associated with therapy resistance and exacerbated tumor cell proliferation [23,31]. PKCα inhibition has an antitumor effect in those types of cancer [72,73]. This was supported by bioinformatic analyses that showed that PKCα is the predominant expressed isoform not only among classical isoforms but also from the whole family of PKC enzymes in those patients, especially in anaplastic TC (ATC), the most aggressive TC subtype. Also, a Kaplan–Meier survival analysis showed that PKCα expression was significantly correlated with poor OS, as previously reported in lung, breast, ovarian, and liver cancer, and high-grade glioma and melanoma, among others [24,31,32,35,37,38,72,73,74]. This correlation was found in male and female TC patients, but men showed a higher decrease in OS than women. Although the incidence of TC is higher in women [1,75,76], it is more aggressive in men (more advanced disease and lower survival rates) [5,77]. Furthermore, TC in men is more likely to present lymph node metastases, tumor size, and lymphatic and vascular invasion, that may affect the initial therapy response or final disease status with a worse prognosis for papillary TC (PTC), especially more advanced disease, and aggressive subtypes [78,79,80]. Interestingly, besides sex-specific malignancies, the ratio of frequency between men and women is >1 for all cancers, except thyroid [81]. It should be further investigated whether high PKCα expression is associated with sex hormones contributing to diminished OS in male TC patients. In addition, our analysis also showed that within patients with low and high TMB, the group with higher PKCα expression has worse OS. This is interesting in the setting of TC where activating mutations are common. That is the case of BRAF mutation in V600E, which occurs in 20–80% of TC cases [82,83]. This mutation is associated with an aggressive phenotype, a high risk of recurrence, and shorter disease-free time compared to BRAF wild-type TC [84,85]. One of the kinases involved in the regulation of signal transduction pathways mediating resistance to therapies targeting BRAFV600E in melanoma is PKCα [38,39,86]; this suggests that a combination therapy targeting both proteins could yield better outcomes in TC patients carrying the BRAFV600E or NRAS mutation, overexpression of PKCα [38]. Given the aforementioned findings, along with the fact that ATC cell lines express high levels of PKCα and harbor BRAF mutations [87], it would be interesting to explore in the future whether PKCα plays a critical role in TC, not only in malignancy but also in therapy resistance. The second most frequent alteration in TC is RAS gene mutation; though it can also occur in both benign and malignant lesions [88], it promotes TC transformation through the activation of the MAPK pathway but exhibits less oncogenic potential than BRAF mutations [89]. Rearrangements involving receptor tyrosine kinases, primarily RET, are present in a subset of thyroid tumors that lack mutations in either BRAF or RAS [90]. Also, mutations in the RET gene have been associated with therapy-acquired resistance [91]. Furthermore, the evaluation of other mutations is becoming increasingly important for prognosis in TC.

Our in silico results were validated by biopsies of PTC patients from our country. Our patient cohort was not particularly large, and this specific subtype of TC is generally less aggressive, so we were not able to find a correlation between PKCα and TC-aggressiveness. However, PKCα staining was primarily detected in adenocarcinoma areas. Tumor expression of PKCα in local patients’ samples together with the higher expression of PKCα in ATC cell lines and our in silico findings let us suggest that PKCα is overexpressed in the more aggressive subtypes of TC. This paves the way for further investigations in this field.

Also, PKCα signaling in TC cell migration was demonstrated [92]; however, the mechanisms by which this isoform acts, particularly in thyroid tissue, remain poorly studied. Thyroid hormones (THs) trigger PKC activation in several cell types through their membrane receptor (mTR), the integrin αvβ3 [93]. The activation of the mTR was widely described to play a central role in cancer cell proliferation, migration, and apoptosis avoidance [94]. Previous studies from our laboratory show that THs can stimulate T-cell lymphoma cell (TCL) proliferation and survival by activating intracellular signals triggered, including PKC via its mTR [49,95]. We confirmed that all the human TC cell lines display high levels of integrin αvβ3 mTR. Furthermore, bioinformatic analysis in public patient databases showed that TC is the cancer with the highest expression of αv and β3 integrin. In accordance, Hoffman and colleagues (2005) showed the presence of integrin αvβ3 in thyroid tumors [96], We found that TH treatment induces in vitro proliferation of different TC types. Furthermore, this effect was inhibited in a concentration-dependent manner by Cilengitide, a selective antagonist of the integrin αvβ3 activity [69,71], highlighting the role of integrin αvβ3 in THs-induced proliferation. The fact that PKCα inhibition impairs THs-induced proliferation would indicate that the intracellular signal involved in THs actions is PKCα. Moreover, knocking down PKCα inhibits AKT and ERK activation, pointing out that both kinases are downstream PKCα. It is to note that TC patients do not show mRNA expression of the other integrins’ (α5, β1, and β5) target of Cilengitide by a bioinformatics analysis in patients with different types of cancer.

Regarding the mechanism involved, Davis et al. showed that the activation of MAPK by T4 through integrin αvβ3 induces glioma cell proliferation, measured by the accumulation of proliferating PCNA, and is dependent on classical PKC [43]. On the other hand, AKT activation through integrin αvβ3 has been widely demonstrated in prostate and lung cancer among others [97,98,99], as well as PI3K/AKT activation by PKC [100,101,102] and THs-induced PKC activation through the mTR [47] integrin αvβ3 [43]. Controversially, it has been shown that PKC inhibitors increase the action of 3,5,3′-triiodo-L-thyronine (T3) on AKT phosphorylation in liver cancer cells [103]. This discrepancy could be explained due to tissue-specific signaling. Our findings are in line with Lin et al. [51] describing that the pharmacological inhibition of integrin αvβ3 decreases the proliferation mediated by the MAPK pathway and sensitizes TC cells to apoptosis.

In line with our in vitro results, our bioinformatic analyses indicated positive correlations between PKCα and PI3K pathways. In this context, we performed an enrichment analysis for genes correlated with PKCα expression that showed the activation of MAPK, PI3K, and cancer-associated pathways in TC patient samples where PKCα is overexpressed.

The present findings demonstrate that PKCα plays a major role in thyroid tumorigenesis, making it a promising therapeutic target and biomarker. A more complete study of these signaling pathways would be necessary, emphasizing the mutations characteristic of TC, such as BRAF, RET/PTC, and RAS [104].

Recently, a PKC theta-targeting inhibitor has entered preclinical development at the clinical stage in metastatic uveal melanoma [105]. Regarding PKCα, different compounds were developed against this kinase [22,106,107,108,109]. It was not possible to assess whether a particular population would benefit from treatment due to the absence of validated predictive biomarkers for drug response during those trials [110]. Furthermore, it is important to consider the role of the isoforms studied in each tumor type. To overcome this common difficulty of drugs targeting signaling pathways which are mediators in cancer, a combination with other drugs to a different target would allow the use of lower doses, thus benefiting from the synergy between both drugs and avoiding the mishaps of high toxicity [111,112].

We found that TC cell lines display several classical, novel, and atypical members of the PKC family and that PKCγ and PKCδ are predominantly expressed in 8505C ATC cells. Few works have studied PKC activity in TC showing proliferative effects [113,114,115,116]. The roles of PKCζ and PKCε have been previously described as promoters of cell proliferation in TC [114]. However, in anaplastic and follicular TC cell lines, PKCβ mediates PMA-induced antiproliferative effect by inducing cell cycle arrest in the G1/S phase [117]. Also, PKCδ has been shown to induce cell cycle arrest and stimulate apoptosis in CT cells [115,117,118]. It is crucial to understand that, when analyzing PKC isoforms as a group, the overall effects observed represent the cumulative actions of individual isoforms. Thus, a more targeted approach is necessary to elucidate the specific roles of each isoform. While this study focuses on PKCα, exploring other isoforms could provide further insights into their contributions. A comprehensive analysis of all PKC isoforms would enhance our understanding of their individual roles. How the different PKC isoforms can be modulated in cancer treatment remains an active area of investigation.

## 4. Materials and Methods

### 4.1. Cell Culture and Treatments

Human-transformed immortalized fetal TAD-2, papillary BCPAP and TPC-1, and anaplastic 8505C, Hth83 and Hth104 thyroid cancer (TC) cells were a kind gift from Judy Meinkoth (University of Pennsylvania, Philadelphia, PA, USA) and were grown in RPMI-1640 (Life technologies, Carlsbad, CA, USA) supplemented with 10% fetal bovine serum (FBS, Gibco, Baltimore, MD, USA). Normal human primary thyroid follicular epithelial Nthy-Ori-3-1 and follicular WRO TC cells were a kind gift from Dr. Guillermo Juvenal (CNEA, Buenos Aires, Argentina) and were grown in the same complete medium. All cells were grown at 37 °C in a humidified 5% CO_2_ incubator.

Cells were arrested by serum starvation for 48 h and then treated with the combination of 100 nM L-thyroxine (T4) and 1 nM triiodothyronine (T3) (Sigma-Aldrich, St. Louis, MO, USA) or 10 mIU/mL bovine thyroid stimulation hormone (bTSH) (Sigma-Aldrich).

### 4.2. Drugs

The following drugs were used in cultures at the final concentrations indicated in Section 2. The protein kinase inhibitor staurosporine (Sigma-Aldrich), the selective PKC inhibitor GF109203X (Bisindolylmaleimide, Tocris Bioscience, Bristol, UK), and the αvβ3 inhibitor Cilengitide (MedKoo, Durham, NC, USA) were dissolved in dimethyl sulfoxide (DMSO, culture grade, Sigma-Aldrich). Stock solutions for each inhibitor were freshly prepared at 1000× in DMSO and control cultures were treated with equal amounts of DMSO (final concentration 0.1–0.3%) in RPMI 1640 medium to achieve the concentrations indicated in Section 2.

### 4.3. Cell Viability Assay

Cell viability was measured with a fluorimetry resazurin reduction test as previously described [119]. Briefly, 5 × 10^3^ cells were plated at a final volume of 0.1 mL in 96-well flat-bottom microtiter plates and were treated with thyroid hormones (THs) for 24 or 48 h. We used Cell Titer-Blue (Promega, Madison, WI, USA), and the fluorescence (560Ex/590Em) was determined using a luminometer (NovoStar microplate reader, BMG Labtech, Ortenberg, Germany). The percentage of viable cells was calculated by the linear least-squares regression of the standard curve. Fluorescence was determined for 6 replicates per treatment condition, and cell viability in hormone-treated cells was normalized to their respective controls.

### 4.4. RNA Interference (siRNA) Transfections

Small interfering RNA (siRNA) SMART Pool: ON TARGET Plus PKCα (siPKCα) and SMART Pool: ON-TARGET Plus Non-Targeting Pool (siNT, control) were purchased from Dharmacon-Thermo Scientific (Lafayette, CO, USA). Three hundred thousand cells were transfected with siRNA sequences (25 nM) using RNAiMax Lipofectamine (Life Technologies) in antibiotic and serum-free medium, and after 30 min the medium was supplemented with 10% FBS. Sixteen hours later, cells were subjected to 48 h of serum starvation followed by various designated treatments and Western blot analysis to check expression levels.

### 4.5. Reverse Transcription (RT) and Quantitative (q) PCR

Reverse transcription and qPCR were carried out as described previously [49]. Briefly, the cell line samples were homogenized in Tri-Reagent (Genbiotech SRL, Buenos Aires, Argentina) and the total RNA was isolated following the manufacturer’s instructions. cDNA was synthesized by retrotranscription using an Omniscript kit (Qiagen GMDH, Germantown, MD, USA). cDNA amounts present in each sample were determined using a commercial master mix for Real-Time PCR containing SYBR Green fluorescent dye (Biodynamics SRL, Buenos Aires, Argentina). qPCR reactions were carried out in an Applied Biosystems 7500 system. Primer sequences (Biodynamics SRL, Buenos Aires, Argentina) were designed using the Primer Express software version 3.0 (Applied Biosystems, Foster City, CA, USA). Quantification of the target gene expression was performed using the comparative cycle threshold (Ct) method according to the manufacturer’s instructions (Applied Biosystems). An average Ct was obtained from the triplicate reactions and normalized to β2-microglobulin, and then ΔΔCt was calculated.

The human primers sequence (5′-3′) was as follows: *PRKCA* fw AAAGGCTGAGGTTGCTGATG; *PRKCA* rv ATTTAGTGTGGAGCGGATGG; *PRKCD* fw CAACTACATGAGCCCCACCT; *PRKCD* rv GGCATTTATGGTGCACATTC; *PRKCE* fw GTCCCTACCTTCTGCGATCA; *PRKCE* rv TCACATCGACGGTGAACATT; *PRKCZ* fw CCAAGAGCCTCCAGTAGACG; *PRKCZ* rv CCATCCATCCCATCGATAAC; *ITGAV* fw AAGTGCCATAGCTCCATTGGGAGA; ITGAV rv TCGAGGATTTGAGATGGCACCGAA; *ITGB3* fw TTCAATGCCACCTGCCTCAACAAC; *ITGB3* rv ACGCACCTTGGCCTCGATACTAAA; *B2M* fw AGATGAGTATGCCGTCCGTGTGAA; *B2M* rv TGCTGCTTACATGTCTCGATCCCA.

### 4.6. Western Blot Analysis

Lysates were prepared from cell lines, as previously described [49]. Equal amounts of proteins (50 μg) were separated by SDS–PAGE on 10% polyacrylamide gels and transferred to nitrocellulose membranes. Nonspecific binding sites were blocked with a blocking buffer (5% nonfat dried milk in PBS 0.1% Tween 20). Membranes were incubated overnight at 4 °C with the following antibodies: PKCα (H-7), PKCε (E-5), PCNA (F-2), TRα,β (FL-408), βactin (C-4), and β-tubulin (F-1) all from Santa Cruz Biotechnology (Santa Cruz, CA, USA); PKCδ, cleaved caspase-3 (Asp175), phospho-Akt (Ser473), and Phospho-p44/42 MAPK (Erk1/2) (Thr202/Tyr204) from Cell Signaling Technologies (Danvers, MA, USA); and PKCβ2, PKCγ, and PKCζ which were from Gibco (Norcross, GA, USA). Secondary mouse or rabbit HRP-conjugated secondary antibodies (Bio-Rad, Hercules, CA, USA) were incubated for 1 h at room temperature. The AmershamTM ECL™ Prime Western blotting detection reagent (GE Healthcare, Piscataway, NJ, USA) was used to develop the protein blot. Blots were developed using ImageQuant LAS 4000 (GE Health Care). Densitometry analysis was performed by the Image J (version 5.1, Silk Scientific Corporation, NIH, Madison, WI, USA) software. Experimental values were referred to as those obtained with the corresponding loading protein band.

### 4.7. Flow Cytometry Analysis

Single-cell suspensions were stained with antibodies against various cell surface markers using standard staining methods as previously described [120]. Briefly, cells were stained with antibodies against the following hormone receptors: TRα/β (sc-772, Santa Cruz Biotechnology) and Integrin αvβ3 (Ab7166, Abcam, Waltham, MA, USA). The following commercially available and fluorochrome-conjugated secondary antibodies were used in the study: PE Goat Anti-Mouse Ig or FITC Goat Anti-Rabbit IgG (BD Biosciences, San Jose, CA, USA). Samples were run on a BD Accury C6 flow cytometer and data were analyzed using the equipment software (BD), version v1.0264.21.

### 4.8. Patients

Patients of both sexes over 18 years of age with papillary TC (PTC), of the classic or follicular variant, were included in the study. Demographic data were extracted from the clinical history and histological preparations were provided by the Pathology Department at the Hospital Británico Buenos Aires, Argentina, and consisted of samples from 9 TC patients. Treatment information and five-year survival status were available for every patient. The patients had a mean follow-up of 32.7 SD 19.9 months, of which 2 were followed for more than 4 years. The study had more females than males, with an average onset age of 41 years. Histopathological analysis was performed by at least two pathologists, and clinicopathological data were collected. Only samples containing >70% tumor tissue were used. The protocol was approved by the ethics committee of the Hospital Británico with the registration number #949. Because of the nature of the study, patients were exempted from signing the informed consent form.

### 4.9. Immunohistochemistry

Immunohistochemistry was performed with the manual method of 4 um-thick, formalin-fixed, paraffin-embedded whole tissue after pressure cooker antigen retrieval (Antigen Unmasking Solution, Tris-Based H-3301, pH 9.0, Vector, St. Batavia, IL, USA) using the anti-PKCα monoclonal antibody (1:100, Santa Cruz Biotechnology SC-8393 H7) and the NovoLink Polymer Detection System (Leica Biosystems, McHenry County, IL, USA) + System-HRP. Coverslips were mounted with DPX Mountant (Sigma-Aldrich). All the experiments were performed at room temperature. Neoplasms that expressed PKCα intensely and diffusely in a cytoplasmic pattern were considered positive. The rest of them were considered negative.

### 4.10. Bioinformatic Analyses

Ethics Statement: All data analyzed in this study are included in public portals that collect datasets that are publicly available.

In silico analysis of patients with different cancer types was performed using different tools, including cBioPortal (www.cbioportal.org, accessed on 30 July and 12 August 2021), an online open-access resource for exploring, visualizing, and analyzing multidimensional cancer genomics data [66,67,121]. For the mRNA expression of ITGAV, ITGB3, ITGA5, ITGB1, and ITGB5 across cancer types, an array of several TCGA-PanCancer Atlas datasets were used in this study (*n* = 2922) [122]. Statistical significance was set at an adjusted *p*-value  < 0.05. The selected genomic profiles contained mRNA expression Z-scores relative to all samples (RNASeq V2 RSEM). The expression levels of target genes were automatically calculated using Z-score ± 2.0, and plots were obtained according to the online instructions in cBioPortal.

The effects of PKCα expression on overall survival (OS) were assessed using the Kaplan–Meier plotter database (http://kmplot.com, accessed on 6 July 2021). The Kaplan–Meier plotter database can assess the effect of 54 k genes on survival in 21 cancer types, including TC (*n* = 502) [123].

We used the R2: Genomics Analysis and Visualization Platform to integrate, analyze, and visualize clinical and genomics data (http://r2.amc.nl, accessed on 26 July 2021). The mRNA expression of all PKC isozymes was analyzed in the GSE126729 dataset (*n* = 28), available on the R2 platform as well as in the Gene Expression Omnibus (GEO) repository.

Metascape (https://metascape.org/gp/index.html, accessed on 12 August 2021) was used to determine pathway and process enrichment in the 750 most significant genes correlated with PKCα expression in those samples from the TCGA PanCancer Atlas TC dataset (*n* = 500) whose PKCα expression was over the median.

### 4.11. Statistical Analysis

One- and two-way ANOVA or Unpaired Student’s *t*-test (two-tailed) analysis was performed using GraphPad Prism (GraphPad Software, La Jolla, CA, USA) version 8. A *p*-value of 0.05 or less was considered significant—* *p* < 0.05, ** *p* < 0.005, *** *p* < 0.0001.

## 5. Conclusions

Here, we show for the first time that protein kinase C alpha (PKCα) is the classical PKC isoform with the highest expression in thyroid cancer (TC) cell lines and in TC cancer patients. Also, we found that Cilengitide treatment reduces thyroid hormones (THs)-induced tumor proliferation. Even if the role of PKC in TH-triggered pathways in the plasma membrane has been previously described in different tissues, including our group, our results highlight for the first time the role of PKCα in THs- and thyroid stimulating hormone (TSH)-mediated proliferation and demonstrate for the first time the importance of the MAPK and AKT activation though PKCα in TSH- and THs-induced proliferation in TC (Figure 7). Furthermore, PKCα expression was significantly correlated with poor OS in TC patients, especially in men. Lastly, PKCα expression was augmented in human thyroid tumors. We can conclude that PKCα overexpression confers an advantage over tumor growth in TC. The use of this protein as a potential biomarker or therapeutic target should be explored in this neoplasm.

## Figures and Tables

**Figure 1 ijms-25-12158-f001:**
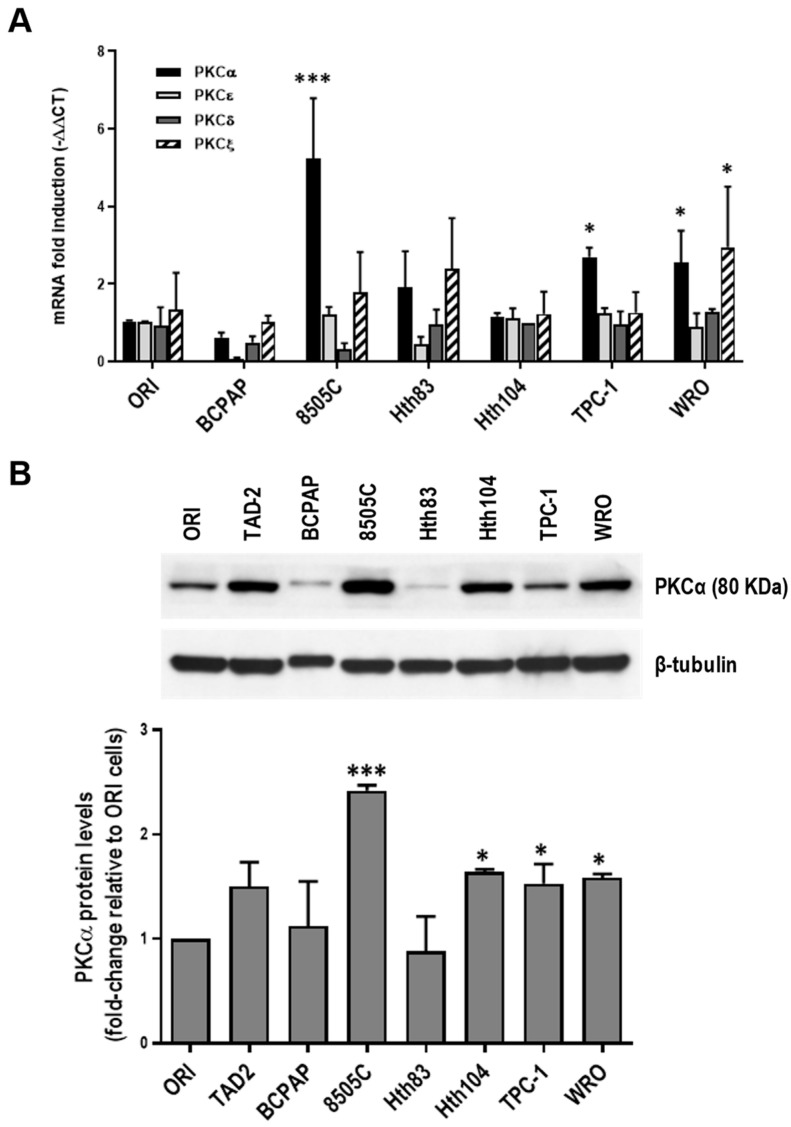
Protein kinase C alpha (PKCα) expression in thyroid cancer (TC) cells. (**A**) mRNA levels of PKCα, PKCδ, PKCε, and PKCζ were analyzed by qPCR in thyroid cells. Gene expression was normalized to the β2-microglobulin gene using the ΔΔCt method. Data are expressed as the mean S.E. of three independent experiments. * *p* < 0.05 and *** *p* < 0.0001 with respect to ORI cells. (**B**) (**Upper panel**) PKCα protein levels in ORI and epithelial TC cells as determined by Western blot analysis. β-tubulin was used as the loading control. Similar results were observed in 3 independent experiments. (**Bottom panel**) Densitometry analysis of PKCα with respect to β-tubulin. * *p* < 0.05: *** *p* < 0.0001 with respect to ORI cells.

**Figure 2 ijms-25-12158-f002:**
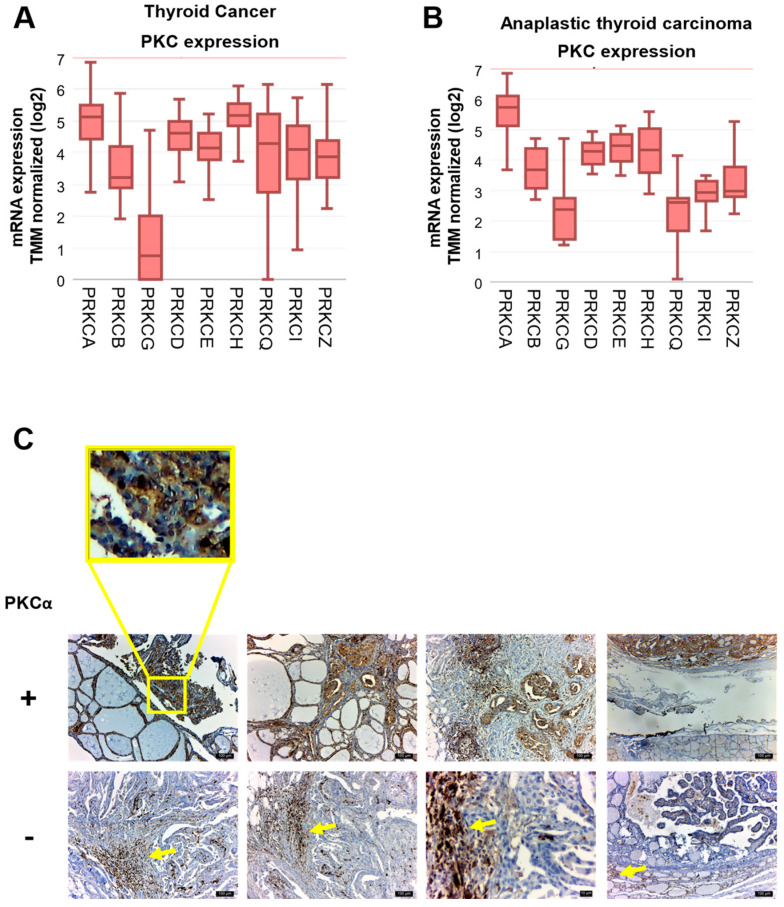
Protein kinase C alpha (PKCα) expression in thyroid cancer (TC) patients. PKC mRNA expression levels of different types of TC were obtained from the Gene Expression Omnibus platform (GSE126729) using R2. (**A**) The expression of all PKC isoforms in TC (papillary PTC + follicular FTC + anaplastic ATC). (**B**) The expression of all PKC isoforms in ATC. (**C**) Representative immunohistochemical staining of PKCα in specimens from TC patients from the British Hospital of Buenos Aires, Argentina. (**Upper panels**), TC with high PKCα expression. (**Bottom panels**), TC with low PKCα expression. (**Inset**), An enhanced view of PKCα staining in a TC specimen.

**Figure 3 ijms-25-12158-f003:**
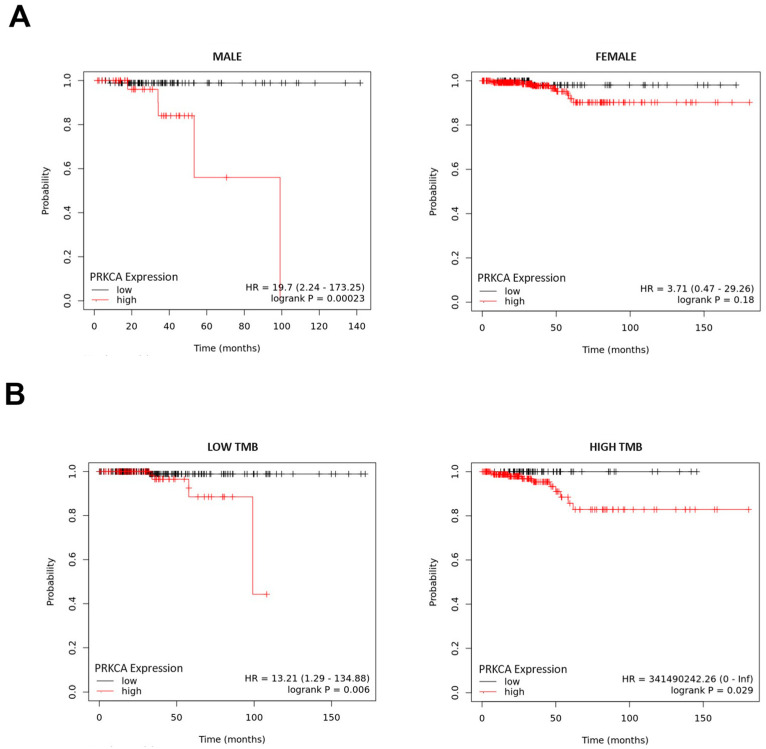
The relationship between protein kinase C alpha (PKCα) expression levels and overall survival in thyroid cancer (TC) patients from a public database. RNA-seq data from the PanCancer Atlas corresponding to TC (TCGA-THCA) were used to analyze for overall survival (OS) through Kaplan–Meier curves. (**A**) (**Left panel**). Male TC patients with high expression of PKCα have a reduced OS than those with low expression (*p* < 0.001). (**Right panel**). Women TC patients with high expression of PKCα have a reduced OS than those with low expression (ns). (**B**) (**Left panel**). Patients with low tumor mutational burden (TMB) and high mRNA expression of PKCα have a reduced OS than those with low expression (*p* < 0.01). (**Right panel**). Patients with high TMB and high mRNA expression of PKCα have a reduced OS than those with low expression (*p* < 0.05). The red curve represents the survival rate of TC patients with high PKCα expression, and the black curve represents the survival rate of TC patients with low PKCα expression.

**Figure 4 ijms-25-12158-f004:**
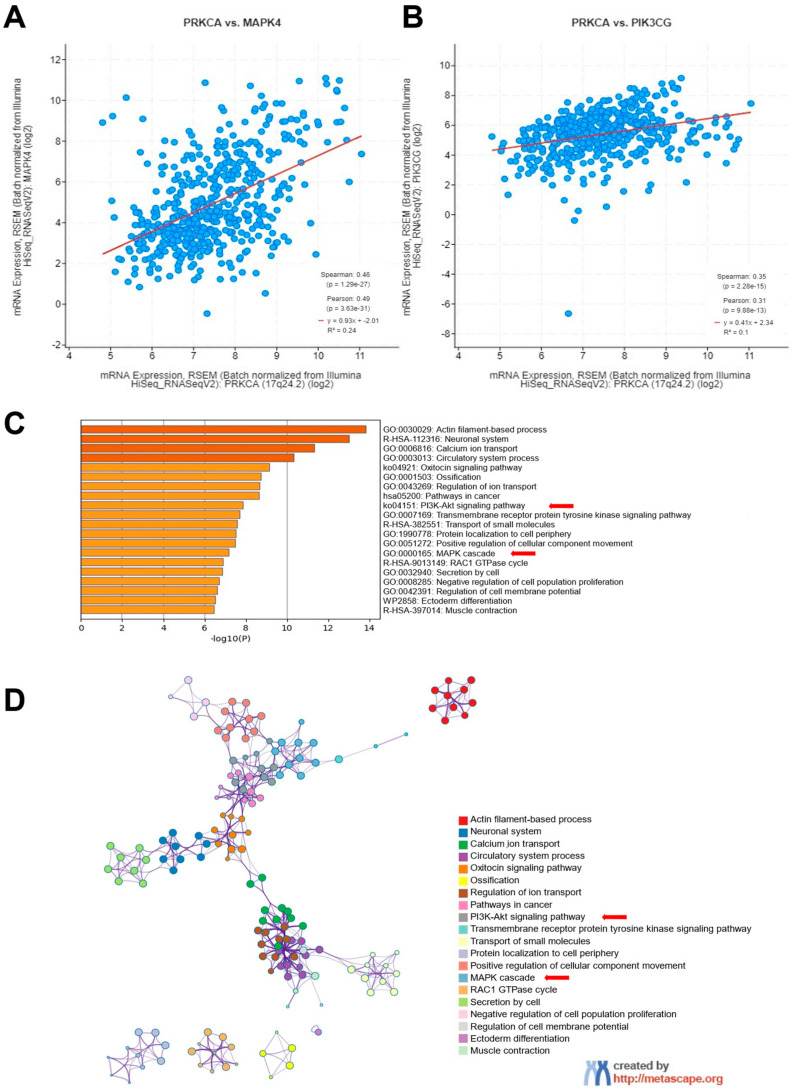
Protein kinase C alpha (PKCα) expression is correlated with MAPK and PI3K pathways in thyroid cancer (TC) patients. Correlation analysis performed on PanCancer Atlas data showed a positive correlation between PKCα and MAPK4 (S = 0.46) (**A**) and PIK3CG (S = 0.35) (**B**) using cBioPortal. (**C**) A bar graph of enriched terms from the top 750 genes most significantly correlated with PKCα in samples with PKCα expression above the median. Cancer-associated signaling pathway functions, including the MAPK cascade and PI3K-AKT signaling, are shown through Metascape analysis. Darker color of the bar associates with a lower p-value. (**D**) A network of enriched terms colored by pathway.

**Figure 5 ijms-25-12158-f005:**
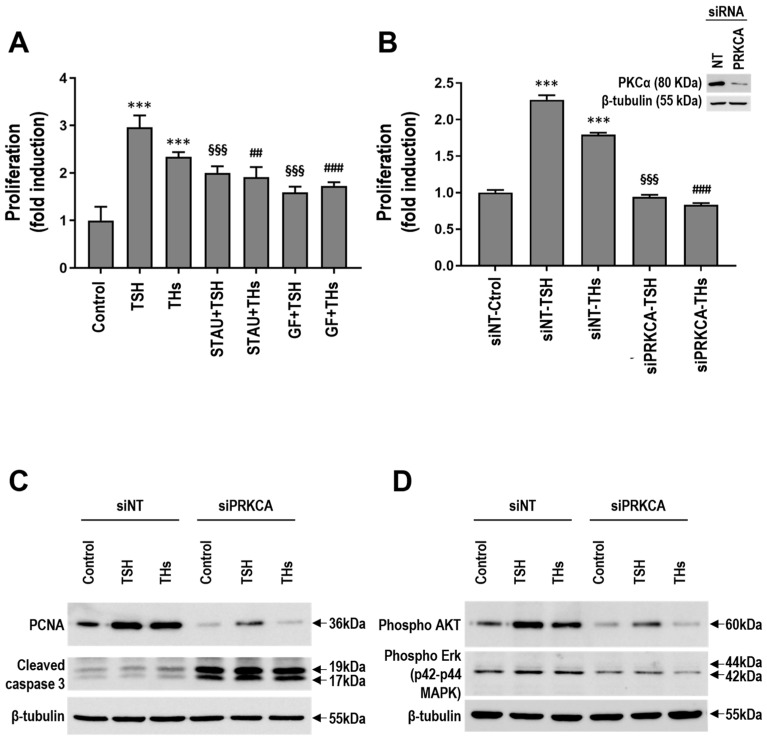
Protein kninase C (PKC)-mediated thyroid stimulating hormone (TSH) and thyroid hormones (THs) effects in thyroid cancer thyroid cancer (TC) cells. (**A**) TPC-1 cells were pretreated or not (Ctrol) with 1 μM staurosporine (STAU) or 5 μM GF109203X (GF) and then incubated with 10 mIU/mL TSH or 1 nM T33,5,3′-triiodo-L-thyronine (T3) and 100 nM L-thyroxine (T4) (physiological concentrations, THs) in combination for 48 h. A Cell Titer Blue assay determined the number of live cells at each dose. Data are shown as the mean ± SD. *** *p* < 0.0001 with respect to the control; §§§ *p* < 0.0001 with respect to TSH; and ## *p* < 0.005 and ### *p* < 0.0001 with respect to TH-treated cells. (**B**) TPC-1 cells were transfected with non-target siRNA (siNT) or PKCα siRNA (si*PRKCA*); the cells were serum starved and untreated (Ctrol) or treated with 10 mIU/mL TSH or 1 nM T3 and 100 nM T4 (physiological concentrations, THs) in combination for 48 h. Cell Titer Blue assay determined the number of live cells at each dose. Data are shown as the mean ± SD. *** *p* < 0.0001 with respect to the control; §§§ *p* < 0.0001 with respect to TSH; and ### *p* < 0.0001 with respect to TH-treated cells. Inset. TPC-1 cells transfected with siNT or si*PRKCA* probed with a PKCα specific antibody. β-tubulin was used as the loading control. Representative data from 1 of 3 independent experiments. (**C**) The expression levels of PCNA and cleaved caspase-3 of cells described in B were determined by Western blot analysis. β-tubulin was used as the loading control. Similar results were observed in 3 independent experiments. (**D**) TPC-1 cells were transfected as in B; the cells were then serum starved and untreated (control) or treated with 10 mIU/mL TSH or 1 nM T3 and 100 nM T4 (THs) for 10 min. Western blot analysis shows AKT and p42/44 MAPK phosphorylation levels. β-tubulin was used as the loading control. Similar results were observed in 3 independent experiments.

**Figure 6 ijms-25-12158-f006:**
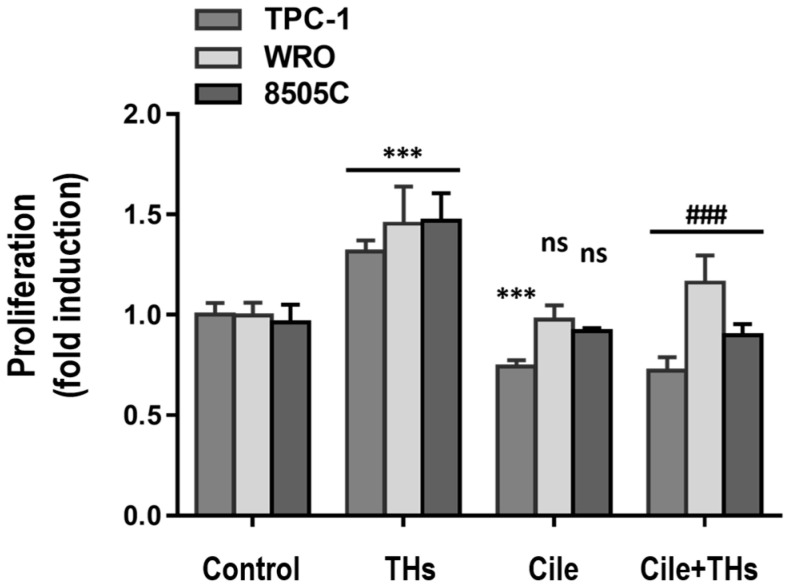
The combined treatment of 3,5,3′-triiodo-L-thyronine (T3) and L-thyroxine (T4) induces the proliferation of thyroid cancer (TC) cells through noncanonical thyroid hormones (THs) receptors. TPC-1, WRO, and 8505C cells were grown in the absence of FBS and then pretreated or not (control) with 4.5 μM Cilengitide (Cile) and then incubated with 1 nM T3 and 100 nM T4 in combination (physiological concentrations, THs) for 48 h by a Cell Titer Blue assay; the number of living cells was determined with each treatment. A commercial kit (Promega) was used as indicated by the manufacturer. The mean ± SD is shown. *** *p* < 0.0001 with respect to the control and ### *p* < 0.0001 with respect to THs and ns means not significant differences with respect to control.

**Figure 7 ijms-25-12158-f007:**
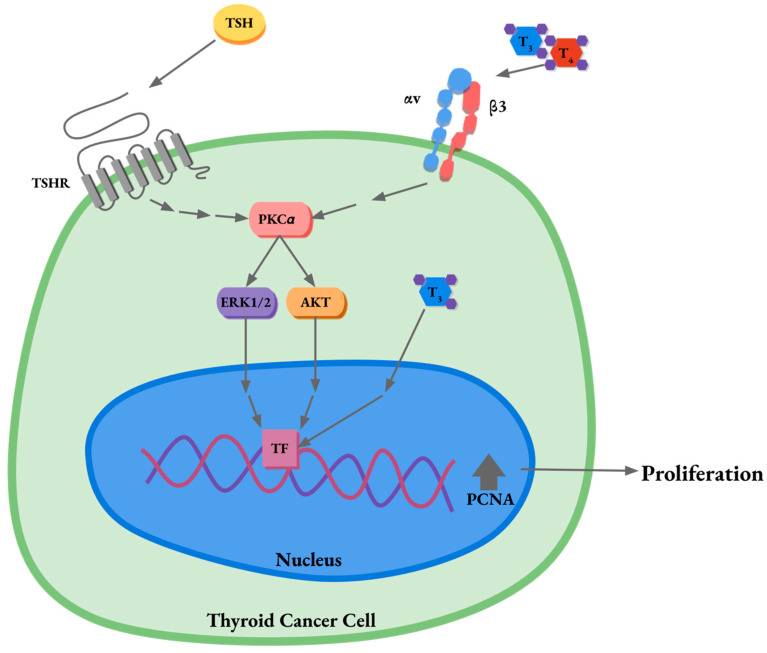
Diagram illustrating protein kinase C alpha (PKCα)-mediated proliferation by thyroid hormones (THs) and thyroid stimulating hormone (TSH) in a thyroid cancer (TC) cell. Proposed mechanisms for PKCα-mediated TH and TSH effects in TC cells (TF: transcription factor).

## Data Availability

All data are available in the authors’ own database.

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
