# Peer review of "PKCα Activation via the Thyroid Hormone Membrane Receptor Is Key to Thyroid Cancer Growth"

_ijms, 2024, doi:10.3390/ijms252212158_

Round 1
Reviewer 1 Report
Comments and Suggestions for Authors
Here, the authors the role of PKCα activation via thyroid hormone membrane receptor in thyroid cancer growth. It highlights that PKCα is overexpressed in thyroid cancer cells and is associated with poor prognosis and therapy resistance. The study found that PKCα mediates hormone-induced proliferation through the integrin αvβ3 receptor, suggesting it as a potential therapeutic target.
Major points
The research provides valuable insights into the molecular mechanisms of thyroid cancer, identifies PKCα as a potential diagnostic and therapeutic target, and uses comprehensive data from both cell lines and patient samples.
The article uses different complementary approaches to reach their conclusion. The bioinformatic and statistical analyses are state of the art.
Minor points
In Figure 5d, the phosphorylated proteins should be normalized to the total Proteins (total AKT, total ERK).
In Figure 2c, the scale bars are missing.
The study is limited by its focus on a single isoform of PKC and may not account for the complexity of thyroid cancer. Additionally, the findings need further validation in clinical settings. This should be discussed in a paragraph citing limitations of the study.
Reviewer 2 Report
Comments and Suggestions for Authors
This study evaluates the functional role of PKCα in the pathogenesis of thyroid carcinogenesis. The study has explored the mechanism and functional aspects of PKCα using cell lines, and patient samples, and utilized cutting-edge molecular technologies. Several following issues must be taken before publication.
Comments
1. Did the author check the expression of other PKC isoforms in cell lines and patient samples?
2. The quality of Figure 2A in Figure 3ABC is blurred and not seen clearly. The resolution should be improved.
3. In Figure 1A, the ORI samples, the bar is missing, and the statistical bar is missing. Either explain and present with a statistical bar.
4. The drugs are dissolved in DMSO. What is the final concentration in the treatment condition? This information should be mentioned in the manuscript.
5. In the western blot of Figure 5 the molecular weight of a specific protein should be shown in the Figure
6. Figures 3A, B, and B are poor and should be replaced with a better resolution.
7. The authors should evaluate the activation of p38 and JNK kinases to find out whether the TSH and THs treatment activates AKT and ERK (Figure 5D) specifically or if it activates other MAPKs as well.
8. BCPAP harbored BRAF and TP53 mutations whereas TPC-1 cells are wild. Is there any correlation of this mutation of functions in this study? Please provide in discussion these points.
Comments on the Quality of English LanguageMinor corrections
Round 2
Reviewer 1 Report
Comments and Suggestions for Authors
The authors addressed all the concerns raised during the first round
Comments on the Quality of English Languageminor typos
Reviewer 2 Report
Comments and Suggestions for Authors
The authors have responded to all comments satisfactorily